# A fraction of *Pueraria tuberosa* extract, rich in antioxidant compounds, alleviates ovariectomized-induced osteoporosis in rats and inhibits growth of breast and ovarian cancer cells

**Swaha Satpathy**[1¤]*, **Arjun Patra**[1], **Muhammad Delwar Hussain**[2]*, **Mohsin Kazi**[3], **Mohammed S. Aldughaim**[4], **Bharti Ahirwar**[1]

**1** Institute of Pharmacy, Guru Ghasidas University, Bilaspur, CG, India, **2** Department of Pharmaceutical & Biomedical Sciences, College of Pharmacy, California Health Sciences University, Clovis, California, United States of America, **3** Department of Pharmaceutics, College of Pharmacy, King Saud University, Riyadh, Saudi Arabia, **4** Research Center, King Fahad Medical City, Riyadh, Saudi Arabia

¤ Current address: Department of Biotechnology & Bioinformatics, Sambalpur University, Burla, Sambalpur, Odisha, India
* swaha22@rediffmail.com (SS); dhussain@chsu.edu (MDH)

## Abstract

*Pueraria tuberosa* (Roxb. ex Willd.) DC., known as Indian Kudzu belongs to family Fabaceae and it is solicited as "Rasayana" drugs in Ayurveda. In the present study, we analyzed the efficacy of an ethyl acetate fraction from the tuber extract of *Pueraria tuberosa* (fraction rich in antioxidant compounds, FRAC) against menopausal osteoporosis, and breast and ovarian cancer cells. The FRAC from *Pueraria tuberosa* was characterized for its phenolic composition (total phenolic and flavonoid amount). Antioxidant property (*in vitro* assays) of the FRAC was also carried out followed by the analysis of the FRAC for its antiosteoporotic and anticancer potentials. The antiosteoporotic activity of FRAC was investigated in ovariectomy-induced osteoporosis in rats. The cytotoxicity effect was determined in breast and ovarian cancer cells. Gas chromatography/mass spectrometry (GC/MS) analysis of the FRAC was performed to determine its various phytoconstituents. Docking analysis was performed to verify the interaction of bioactive molecules with estrogen receptors (ERs). The FRAC significantly improved various biomechanical and biochemical parameters in a dose-dependent manner in the ovariectomized rats. FRAC also controlled the increased body weight and decreased uterus weight following ovariectomy in rats. Histopathology of the femur demonstrated the restoration of typical bone structure and trabecular width in ovariectomized animals after treatment with FRAC and raloxifene. The FRAC also exhibited *in vitro* cytotoxicity in the breast (MCF-7 and MDA-MB-231) and ovarian (SKOV-3) cancer cells. Furthermore, genistein and daidzein exhibited a high affinity towards both estrogen receptors (α and β) in the docking study revealing the probable mechanism of the antiosteoporotic activity. GC/MS analysis confirmed the presence of other bioactive molecules such as stigmasterol, β-sitosterol, and stigmasta-3,5-dien-7-one. The FRAC from *Pueraria tuberosa*

**Data Availability Statement:** All relevant data are within the paper and its Supporting Information files.

**Funding:** The authors deeply acknowledge University Grants Commission, New Delhi, India, for financial support as Raman Fellowship [F.NO.5-63/2016(IC)] to AP.

**Competing interests:** The authors have declared that no competing interests exist.

has potential for treatment of menopausal osteoporosis. Also, the FRAC possesses anticancer activity.

## Introduction

The World Health Organization (WHO) defines osteoporosis as a decrease of bone mineral density (BMD) more than 2.5 standard deviations of the standard reference for BMD in young healthy women [1]. Osteoporosis deteriorates BMD, and bone architectural structure. It also enhances the risk of fracture. Besides, osteoporosis causes severe problems to the quality of life, such as disability, and even death [2]. The cause of osteoporosis is variation in bone-forming (osteoblastic) and bone-resorbing (osteoclastic) cell function [3]. Osteoporosis has the highest prevalence in senile people and severely affects about 50% of menopausal women worldwide. The number of people with age ≥60 years in India is estimated to be 315 million by 2050 from 35 million in 2013, signifying the possibility of higher incidence of osteoporosis [4]. A decrease in the level of estrogen is the key contributing feature for menopausal osteoporosis (MO) in women. The reduced estrogen causes diminished bone formation, enhanced bone resorption, and elevated production of proinflammatory cytokines such as IL-1, IL-6, IL-7, and TNF-α [5]. The occurrence of MO is increasing day by day because of inactive lifestyle, environmental vulnerability, amenorrhea, hormonal alterations, early inception of puberty, and ovarian disorders [6, 7]. Furthermore, several studies have demonstrated oxidative stress prevalent in MO declines the antioxidant defense, and these lowered antioxidant levels promote bone loss [8, 9]. Oxidative stress can reduce the life span of osteoblasts by inhibiting osteoblastic differentiation and promoting bone resorption by boosting the development and activity of osteoclasts, thus causing osteoporosis. In MO, the activated osteoclasts produce reactive oxygen species like superoxides and a rise in the malondialdehyde level in blood. These oxidative stresses also contribute to bone loss in osteoporosis [4]. Antioxidants can be useful in the management of MO by normalizing the altered osteoblastic and osteoclastic functions [10].

Several drugs, such as estrogens, biphosphonates, and parathyroid hormone analogs are used for the inhibition and management of osteoporosis. They promote bone formation or decrease bone resorption or both [11]. However, these treatments are characterized by serious concerns related to their safety and efficacy. Estrogen therapy is not preferred in patients with hepatopathy and venous embolism. Also, there is high possibility of developing breast, cervical, and ovary cancer, development of heart disease and stroke upon long-term use of estrogen [12]. Long-term application of biphosphonates shows adverse effects such as osteonecrosis of the jaw and atypical femoral fractures [13]. Parathyroid hormone analogs are however costly, and with patients needing daily injection, the analogs may cause adverse consequences like osteosarcoma [14]. Therefore, it is important to develop drugs from plant origin that has a protective effect on bone loss with fewer side effects. Plant-derived estrogenic compounds are known as "Phytoestrogens" and are accepted worldwide as safe treatments for MO [7]. The phytoestrogens mostly include isoflavones, isoflavanones, coumestans, flavanones, chalcones, and flavones [15]. A considerable number of plant drugs in the form of extracts, fractions, herbal preparations, and isolated molecules have been studied to prevent or control osteoporosis [16]. Although these plant-derived remedies are helpful in the management of MO, they may produce the side effects of supplemental estrogen [17, 18]. Hence, a search for safe, cheap, and effective plant-derived agents for the management of MO is required.

Different species of *Pueraria* such as *Pueraria lobata*, *Pueraria mirifica*, and *Pueraria candollei* var. mirifica have been studied as protective agents against bone loss [19–21]. *Pueraria*

*tuberosa*, known as Indian Kudzu belongs to the family Fabaceae, is used as "Rasayana" drugs in Ayurveda. It is used in various Ayurvedic preparations, traditional management of a wide range of ailments, and studied for an array of pharmacological activities. The plant is a rich source of various secondary metabolites and contains phytoestrogenic compounds such as puerarin, quercetin, genistein, and daidzein [22]. Despite the significant pharmacological and phytochemical potential, the antiosteoporotic activity of *Pueraria tuberosa* has not been explored. Our objective was to identify a fraction rich in antioxidant compounds (FRAC) from the tubers of the plant and investigate the antiosteoporotic and anticancer effect of the FRAC.

## Materials and methods

### Chemicals, reagents and kits

The following high grade chemicals were obtained commercially or as a generous gift: 3-(4,5-dimethylthiazol-2-yl)-2,5-diphenyltetrazoliumbromide (MTT) (Sigma-Aldrich, St Louis, MO, USA); raloxifene (Cipla Ltd., Goa, India); phosphorous, calcium, alkaline phosphatase (ALP), tartrate-resistant acid phosphatase (TRAP), hydroxyproline (HP), total cholesterol (TC) and triglyceride (TG) kits (Span Diagnostic Pvt. Ltd.); dimethyl sulfoxide (DMSO) and phosphate buffer saline (PBS) (Mediatech Inc., Manassas, VA, USA); xylazine (Indian Immunologicals Ltd., Hyderabad, India); ketamine (Neon Laboratories Limited, Thane, India); diclofenac (Troikaa Pharmaceuticals Ltd., Ahmedabad, India); gentamicin (Abbott, Pitampur, India); DPPH (1,1-diphenyl-2-picrylhydrazyl) (HIMEDIA Co. Ltd., India); ABTS (2,2'-azino-bis(3-ethylbenzothiazoline-6-sulfonic acid) assay kit (Sigma-Aldrich, MO, USA, Catalog Number MAK187); OxiSelect™ TAC Assay Kit (Cell Biolabs, Inc., San Diego, CA, USA; Catalog Number: STA- 360); fetal bovine serum (Mediatech, Manassas, VA); penicillin streptomycin solution 100X (10,000 IU/mL penicillin and 10,000 μg/mL streptomycin) (Mediatech, Manassas, VA) were procured.

### Extraction of plant material and fractionation

Tubers of *Pueraria tuberosa* were collected from Bilaspur, Chhattisgarh, India, with the help of the traditional practitioners and authenticated through the ICAR-National Bureau of Plant Genetic Resources, Regional Station, Phagli, Shimla, India (No.: NBPGR-565-569). A voucher specimen has been preserved in the Institute of Pharmacy, GGU, Bilaspur for future references. The fresh tubers were cut into small pieces and dried under shade, then coarsely powdered and stored in an air-tight container until further use. The coarse powder material was extracted with ethanol using soxhlet apparatus. The extract was concentrated under reduced pressure using a rotary vacuum evaporator to obtain the mother extract (concentrated ethanol extract, PT). The concentrated extract was suspended in distilled water and successively fractionated by liquid-liquid partitioning with n-hexane, ethyl acetate and n-butanol. Finally, the remaining aqueous fraction was also prepared. All the fractions were dried and stored in an airtight container until further use.

### Identification of fraction rich in antioxidant compounds (FRAC)

The mother extract (PT), n-hexane fraction (PT1), ethyl acetate fraction (PT2), n-butanol fraction (PT3) and aqueous fraction (PT4) were evaluated for the antioxidant potential (by DPPH assay, ABTS assay and finding total antioxidant capacity) and phenolic composition (by total phenolic and flavonoid content determination) to identify the best fraction rich in antioxidant compounds (FRAC).

**DPPH assay.** Scavenging of 1,1-diphenyl-2-picrylhydrazyl (DPPH) free radical by all the samples was measured spectrophotometrically [23]. Two milliliters of the samples of different concentrations were added to one milliliter of 0.2 mM DPPH solution prepared in methanol. Methanol was used as a control in place of the samples. The solutions were kept at room temperature for one hour in the dark, and then the absorbance was read at 517 nm. The potential of free radical scavenging was represented as the percentage inhibition of DPPH free radical and was calculated using the following formula. The concentration of samples producing 50% inhibition ($IC_{50}$) was also determined.

$$\%Inhibition = \frac{(C - S)}{C} X\ 100$$

Where, C = absorbance of the control and S = absorbance of the sample.

**ABTS assay.** The antioxidant capacity of the samples was analyzed based on their ability to interact with ABTS radicals [24]. The assay was performed following the protocol provided with the assay kit from Sigma-Aldrich, MO, USA (Catalog Number MAK187). The kit components were $Cu^{+2}$ reagent (Catalog Number MAK187A), assay diluent (Catalog Number MAK187B), protein mask (Catalog Number MAK187C), and Trolox standard, 1.0 μmole (Catalog Number MAK187D). Briefly, 10 μL of the sample, 90 μL of HPLC water, and 100 μL of $Cu^{+2}$ working solution were transferred to each well in a 96 well plate. The contents were mixed thoroughly using a horizontal shaker and incubated in light protected condition at room temperature for 90 min. Finally, the absorbance was read at 570 nm, and the Trolox equivalent as μM/g of the sample was determined from the standard curve of Trolox.

**Determination of total antioxidant capacity (TAC).** TAC, in terms of copper reducing equivalent (CRE) of the sample, was evaluated using OxiSelect™ TAC Assay Kit (Cell Biolabs, Inc., San Diego, CA, USA; Catalog Number: STA- 360) [25]. The components of the kit were uric acid standard (Part No. 236001), reaction buffer, 100X (Part No. 236002), copper ion reagent 100X (Part No. 236003) and stop solution, 10X (Part No. 236004). The assay protocol was as per the manufacturer's product manual. Briefly, 20 μL of the sample in various concentrations and 180 μL of 1X reaction buffer were transferred to each well in a 96 well plate and mixed thoroughly. An initial absorbance was taken at 490 nm. The reaction was started by adding 50 μL of 1X copper ion reagent into each well and incubated on an orbital shaker for 5 min. Then the reaction was stopped by adding 50 μL of 1X stop solution to each well and absorbance was measured again. The net absorbance was calculated by subtracting the initial reading from the final reading and the mM uric acid equivalent (UAE) was determined from the uric acid standard curve. Finally, the CRE was determined by multiplying UAE by 2189.

**Determination of total phenolic and flavonoid content.** Folin-Ciocalteu method and aluminum chloride colorimetric method were adopted for determining total phenolic content (TPC) and total flavonoid content (TFC), respectively [26] by reconstituting the samples in methanol. For determination of TPC, 100 μL of the sample (1.0 mg/mL) was mixed with 125 μL of Folin-Ciocalteu reagent and 750 μL of sodium carbonate solution (15% w/v) in a test tube. The final volume was adjusted to 5 mL with deionized water and mixed properly. The mixture was incubated at room temperature in the dark for 90 min, and then the absorbance was read at 760 nm using a spectrophotometer (UV-1800, Shimadzu, Japan). A blank sample with water and reagents were prepared and used as a reference. TPC of the samples was represented as milligrams of gallic acid equivalents per gram dry weight (mg of GAE/g DW) of a sample through the calibration curve of gallic acid. For TFC estimation, 0.5 mL of sample (0.1 mg/mL) was mixed with 0.1 mL of $AlCl_3$ (10%), 0.1 mL of potassium acetate (1 molar) and 1.5 mL of methanol (95%). The final volume was adjusted to 5 mL with distilled water and mixed thoroughly. The mixture was incubated in dark at room temperature for 60 min and then

absorbance was measured at 415 nm. TFC was expressed as mg of rutin equivalents per gram (mg RE/g) of the sample through a standard curve of rutin. All measurements were carried out in triplicate.

## Gas chromatography mass spectrometry (GC/MS) analysis of the FRAC

Ethyl acetate fraction was identified as the fraction rich in antioxidant compounds (FRAC). GC/MS analysis was carried out on a GC/MS system comprising of Thermo Tracer 1300 GC and Thermo TSQ 8000 MS. The GC was connected to an MS with the following conditions such as TG 5MS (30m X 0.25 mm X 0.25 μm) column, operating in electron impact [electron ionization positive (EI+)] mode at 70 eV, helium (99.999%) as carrier gas at a constant flow of 1 ml/min, S/SL injector, an injection volume of 1.0 μl (split ratio of 10:1), injection temperature 250°C and MS transfer line temperature 280°C. The oven temperature was programmed from 60°C (isothermal for 2 min), with a gradual increase in steps of 10°C/min to 280°C. Mass spectra were taken at 70 eV, a scanning interval of 0.5 sec, and a full mass scan range from 50 m/z to 700 m/z. Data acquisition was carried out by Xcalibur 2.2 SP1 data acquisition software. Interpretation of the mass spectrum of GC/MS was performed by the NIST (National Institute Standard and Technology) mass spectral search program for the NIST/EPA/NIH mass spectral library version 2.0 g. NIST 11. The mass spectrum was compared with the spectrum of the components stored in the NIST library. The chemical name, molecular formula, and molecular weight of the compounds were determined.

## Antiosteoporotic activity

**Animals.** Virgin female Wistar rats weighing 220–250 g were housed in polypropylene cages (two per cage) in an air-conditioned room at 23±1°C, the relative humidity of 50–60% and 12 h/12 h light/dark illumination cycle. The animals were provided free access to diet and water. The animal use and experimental protocol (Reference No.: 119/IAEC/Pharmacy/2015) was approved by the Institutional Animal Ethical Committee (IAEC) of the Institute of Pharmacy, Guru Ghasidas University, Bilaspur, Chhattisgarh (Reg. No.: 994/GO/Ere/S/06/CPCSEA) under the guidelines of CPCSEA.

**Acute oral toxicity study.** An OECD (The Organization for Economic Co-operation and Development) 423 guideline was employed to determine the acute oral toxicity of FRAC. The limit test was performed as per the guidelines on female rats (three rats per step) at a dose of 2000 mg/kg, orally and monitored for 14 days. The FRAC was suspended in carboxy methylcellulose (1.0%). No mortality or any signs of moribund status were found at this dose (2000 mg/kg). Therefore, the $LD_{50}$ cut-off is 5000 mg/kg (category 5 in the Globally Harmonized Classification System). The dosages selected for the antiosteoporotic property were 100 and 200 mg/kg/day. Usually, 1/5th and 1/10th of the $LD_{50}$ value can be used for animal experimentation and the dose selected (very small dose compared to $LD_{50}$ cut-off, 5000 mg/kg) for screening antiosteoporotic activity, where the treatment was for 90 days. We have not observed and mortality of signs of toxicity during this 90 days period also.

**Experimental protocol.** The animals were acclimatized for seven days. On the seventh day, rats were ovariectomized and sham-operated after anesthetized with intraperitoneal administration of ketamine and xylazine. For ovariectomized (OVX) rats, the ovaries were bilaterally removed by a small midline skin incision. In the case of sham-operated rats, the ovaries were exposed and sutured back without removing them [27]. Postoperative care was taken by administering diclofenac and gentamicin, and individual housing of the animals for a few days. After four weeks, the animals were divided into five groups, each group containing six animals and received following treatment for 90 days.

Group I: Sham-operated and received 1% CMC (Sham control).

Group II: Ovariectomized animals and received 1% CMC (OVX control).

Group III: Ovariectomized animals treated with standard drug, raloxifene (1 mg/kg) (RAL).

Group IV: Ovariectomized animals treated with FRAC (100 mg/kg) (FRAC-100).

Group V: Ovariectomized animals treated with FRAC (200 mg/kg) (FRAC-200).

At the end of treatment, food was withheld from the animals for 24 h, and then urine samples were collected in metabolic cages. Urine samples were refrigerated until further investigation. Animals were euthanized by ether anesthesia, and blood samples were immediately withdrawn from the abdominal aorta. The blood samples were centrifuged at 2500 rpm for 25 min, serums were separated, and stored at -70˚C until analysis. The uterus was taken out carefully after the blood withdrawal and weighed. The femur and fourth lumbar vertebrae were collected by detaching the connecting tissue and stored at -70˚C until the biomechanical parameters were determined.

**Determination of biochemical parameters.** Various serum parameters were determined by using commercial assay kits. The parameters include calcium, phosphorus, alkaline phosphatase (ALP), tartrate-resistant acid phosphatase (TRAP), triglycerides (TG), and total cholesterol (TC). Hydroxyproline (HP), calcium, and phosphorus in urine were also determined as reported earlier [4, 11].

**Determination of biomechanical parameters.** Weight (by digital balance, CY 204, Citizon, India), length (between the proximal tip of the femur head and the distal tip of medial condyle), and thickness (using Vernier caliper) of the femurs were measured after drying overnight and removal of bone marrow. Bone volume (by plethysmometer, INCO Instruments & Chemicals Pvt. Ltd., Ambala, India) and bone density (mass/volume) were also determined. The breaking strength of the femur and fourth lumbar vertebrae were evaluated using a hardness tester [11, 28].

**Determination of body weight and organ weight.** Bodyweight of each animal was measured on the first day and the last day of treatment. Uterus weight was also measured immediately after its removal and detachment of uterine horns, fat, and connective tissues [11, 29]. The uterus weight relative to body weight was also calculated.

**Histopathology of the femur.** The right femur was fixed in 10% formalin for 12 h at 4˚C, decalcified in ethylenediamine tetraacetic acid (EDTA) for 7 days, dehydrated, defatted, embedded in paraffin wax, and sectioned in the sagittal plane (5 μm thickness) using a microtome. The sections were stained with hematoxylin and eosin (H & E) and examined for histopathological changes under a light microscope (Primo Star, Zeiss with AxioCam ERc 5s camera) [7].

### *In vitro* cytotoxicity of fraction rich in antioxidant compounds (FRAC)

Breast (MCF-7 and MDA-MB-231) and ovarian (SKOV-3) cancer cells were cultured in Dulbecco's Modified Eagle's Medium (DMEM) supplemented with 10% fetal bovine serum and 1% penicillin/streptomycin. The cells were harvested and transferred to 96-well plates at a density of 3000 cells/well. The plates were incubated at 37˚C and 5% $CO_2$. After 24 h, the medium was replaced with fresh medium containing FRAC at various concentrations (31.5 to 500 μg/mL). The culture medium without any drug formulation was used as the control. After 72 h of incubation at 37˚C and 5% $CO_2$, the media was removed carefully, and cells were washed twice with sterile phosphate buffer saline (PBS). 50 μl of MTT solution (0.5 mg/ml in DMEM media) was put into each well and further incubated for 4.0 h in the same condition. The

medium was then removed from each well, and 100 μl of DMSO was added to each well to dissolve the purple formazan crystal obtained from the MTT assay. The absorbance value of each well was read at 570 nm using a microplate reader (Varioskan Flash, Thermo Scientific, USA). The percent cell viability with different treatments was calculated from the following formula [7, 30].

$$\%\text{Cell Viability} = \frac{Absorbance\ of\ Test}{Absorbance\ of\ Control} X\ 100$$

**Docking study of the identified phytoconstituents in FRAC.** In our earlier study, we have reported the presence of genistein and daidzein in the fraction rich in antioxidant compounds (the ethyl acetate fraction) of *Pueraria tuberosa* [31]. A docking study of these two compounds with estrogen receptor α (1x76) and estrogen receptor β (1x7R) [ER-α and ER-β] was performed to elucidate the mode of their interaction. The values of the grid box (x, y and z) are, 1x7R: 15.587, 32.224, 22.304 and 1x76: 29.554, 37.747, 38.951. All computational studies were carried out using FlexX LeadIT 2.1.8 of BiosolveIT in a Machine running on a 2.4 GHz Intel Core i5-2430M processor with 4GB RAM and 500 GB Hard Disk with Windows 10 as the Operating System. The 3D conformer of the ligands was downloaded from PubChem in. sdf format. Reference protein coordinates of ER-α and ER-β for docking studies were obtained from X-ray structures deposited in Protein Data Bank (http://www.rcsb.org). For protein preparation, the chain having the receptor was selected as receptor components. Then reference ligand was selected. 3D conformer (.sdf) of daidzein and genistein were downloaded from PubChem website and used as ligand. FlexX uses only 3D conformer of ligands. All the chemical ambiguities, which were crystallographically unresolved structures, were resolved, and the receptor was confirmed. The docking process deals with the translational, torsional, and ring conformation degrees of freedom. It was done by "Define Flex Docking" utility, and the FlexX accurately predicted the geometry of the protein-ligand complex within a few seconds. Then the docking was done using default parameters using a hybrid approach, followed by visualization using Pose View. The best conformation for each ligand sorted by the final binding affinity was stored [32].

## ADME prediction

ADME (Absorption, Distribution, Metabolism and Excretion) is a key aspect to predict the pharmacodynamics of the molecule under study which could be used as a future lead molecule for drug development. SWISSADME is an online web server developed and maintained by the Molecular Modeling Group of the Swiss Institute of Bioinformatics (SIB) (https://www.swissadme.ch) (SIB, 2019). We have evaluated the ADME profiles of both the compounds under study as they showed good interaction with estrogen receptors. To compute ADME parameters, already prepared structures of ligands/molecules were uploaded individually in Marvin JS section provided in the website, http://swissadme.ch/index.php. Structures were auto converted to SMILES format and then ADME was predicted by the server. Results obtained were saved for further analysis.

## Toxicity prediction

Prediction of toxicity is an important aspect for all molecules. The pkCSM is a web server database in which analysis of molecules are done by drawing the small molecule virtually or by submitting the SMILES format of the same. The web server database (http://biosig.unimelb.edu.au/pkcsm/prediction) provides details of toxicity namely AMES toxicity, maximum tolerated

**Table 1. Total phenolic and flavonoid content, and antioxidant potential of ethanol extract and different fractions of *Pueraria tuberosa*.**

| Sample | TPC (mg GAE/g DW) | TFC (mg RE/g DW) | IC$_{50}$ (DPPH method) (μg/mL) | μM Trolox equivalent/g sample (ABTS assay) |
|---|---|---|---|---|
| PT | 12.0 ± 0.70 | 40.26 ± 0.83 | 597.5 ± 7.89 | 376.13 ± 8.72 |
| PT1 | 0.95 ± 0.07 | 2.19 ± 0.35 | 1396.72 ± 15.85 | 45.87 ± 3.79 |
| PT2 | 106.23 ± 1.66 | 261.9 ± 1.73 | 55.70 ± 3.15 | 907.51 ± 8.07 |
| PT3 | 56.0 ± 1.30 | 25.56 ± 0.65 | 110.27 ± 10.41 | 360.26 ± 8.35 |
| PT4 | 26.46 ± 1.66 | 12.67 ± 0.77 | 291.08 ± 6.33 | 92.10 ± 4.84 |

Values are mean ± SEMs (n = 3). PT, ethanol extract of *Pueraria tuberosa*; PT1, n-hexane fraction; PT2, ethyl acetate fraction; PT3, n-butanol fraction; PT4, aqueous fraction; TPC, total phenolic content; TFC, total flavonoid content; GAE, gallic acid equivalent; DW, dry weight; RE, rutin equivalent; IC$_{50}$, the concentration that provides a reduction of 50%; DPPH, 1,1-diphenyl-2-picrylhydrazyl; ABTS, 2,2'-azino-bis(3-ethylbenzothiazoline-6-sulfonic acid).

dose, hepatotoxicity, skin sensitization, hERG I and II inhibitor. The website was logged on and the SMILES of both the molecules were searched and submitted into the website and toxicity was selected in prediction mode [33]. Finally, results were obtained.

## Statistical analyses

Data were presented as mean ± standard error means (SEMs). The data obtained in antiosteoporotic activity were subjected to one-way analysis of variance (ANOVA) followed by Newman-Keuls multiple comparisons for significance using GraphPad Prism 7.0 (GraphPad Software, La Jolla, CA, USA) software. A value of $p < 0.05$ was considered statistically significant.

## Results

### Characterization of the fraction rich in antioxidant compounds

The antioxidant potential of different samples (ethanol extract, and n-hexane, ethyl acetate, n-butanol and aqueous fractions) was evaluated based on their phenolic composition (total phenolic and flavonoid content as gallic acid equivalent and rutin equivalent, respectively), and antioxidant potential (by DPPH method, ABTS assay and determining total antioxidant capacity). The ethyl acetate fraction (PT2) contained maximum phenolics and antioxidant activity (Tables 1 and 2), and chosen as the fraction rich in antioxidant compounds (FRAC). This fraction was further analyzed for antiosteoporotic and anticancer activities.

### GC/MS analysis of FRAC

FRAC from *Pueraria tuberosa* contained 23 different chemical moieties (S1 Table) including stigmasterol, β-sitosterol, and stigmasta-3,5-dien-7-one.

**Table 2. Copper reducing equivalent of ethanol extract and different fractions of *Pueraria tuberosa* at different concentrations.**

| Concentration (μg/mL) | Copper reducing equivalent | | | | |
|---|---|---|---|---|---|
| | PT | PT1 | PT2 | PT3 | PT4 |
| 12.5 | 4.38±0.10 | 2.19±0.04 | 6.57±0.30 | 6.79±0.62 | 5.47±0.08 |
| 25 | 5.47±0.18 | 4.38±0.07 | 21.89±0.62 | 9.41±0.30 | 5.91±0.04 |
| 50 | 6.57±0.22 | 5.47±0.11 | 35.02±0.48 | 9.85±0.28 | 6.57±0.12 |
| 100 | 8.76±0.12 | 9.85±0.22 | 72.24±2.12 | 10.07±0.80 | 9.85±0.34 |
| 200 | 10.95±0.28 | 15.32±0.56 | 113.83±1.62 | 26.27±0.74 | 13.13±0.82 |

Values are mean ± SEMs (n = 3); PT, ethanol extract; PT1, n-hexane fraction; PT2, ethyl acetate fraction; PT3, n-butanol fraction; PT4, aqueous fraction.

## Antiosteoporotic activity

The antiosteoporotic potential of FRAC of *Pueraria tuberosa* was evaluated in ovariectomized-induced osteoporosis in female rats by determining the following parameters:

**Effect of FRAC on biochemical parameters in serum and urine.** Both serum and urine were analyzed for the levels of phosphorous (P) and calcium (Ca) (Figs 1 and 2). In the OVX control group, as well as in other treatments, there was no significantly change in serum P and Ca. The level of P and Ca in urine increased significantly in the OVX group over sham control. Administration of both doses of FRAC and raloxifene significantly reduced (p<0.05) the OVX-induced increase in urine levels of P and Ca. Levels of bone markers, ALP and TRAP were significantly enhanced (p<0.001) after OVX. Both ALP and TRAP levels were reduced significantly (p<0.001) and dose-dependently after FRAC treatment (versus OVX). Serum ALP and TRAP level were also reduced significantly (p<0.001) after raloxifene treatment (Fig 1). The OVX caused a significant increase (p<0.001) in the level of urine hydroxyproline (HP) compared to sham control. However, the level of HP in raloxifene and FRAC (100 and 200 mg/kg) treated groups were distinctly lowered (p<0.001) compared to the OVX group (Fig 2). The level of TC and TG increased significantly (p<0.001) in the OVX group compared to the sham control group (Fig 1). These increased TC and TG level was markedly lowered (p<0.01) by FRAC and raloxifene treatment. TG levels in raloxifene and FRAC-200 groups are comparable with the sham control, and FRAC exhibited a better effect over raloxifene.

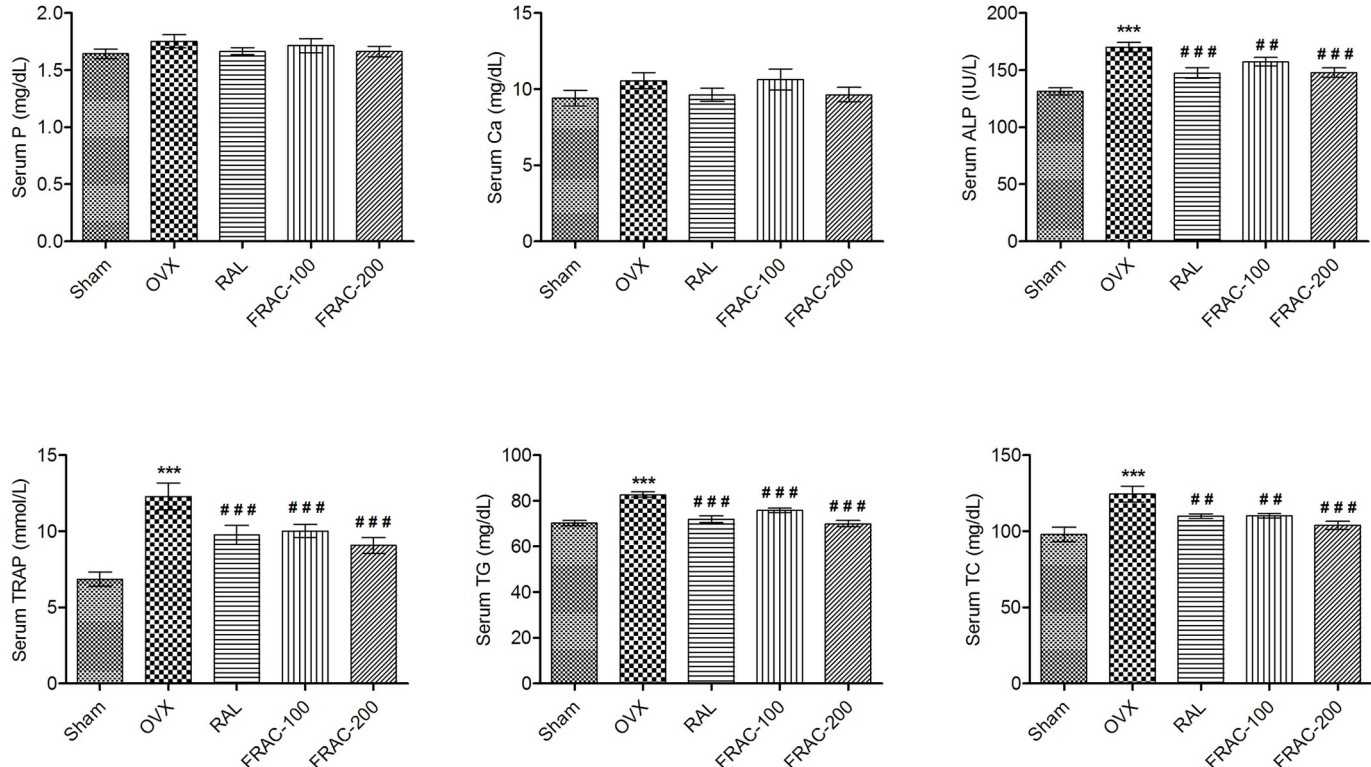

**Fig 1. Effect of FRAC from *Pueraria tuberosa* on biochemical parameters of serum.** Data were average ± SEM (n = 6). *** p < 0.001 significantly different from sham control group. ## p < 0.01, ### p < 0.001 significantly different from OVX group. Ca, calcium; P, phosphorus; ALP, alkaline phosphatase; TRAP, tartrate resistant acid phosphatase; TG, triglycerides; TC, total cholesterol.

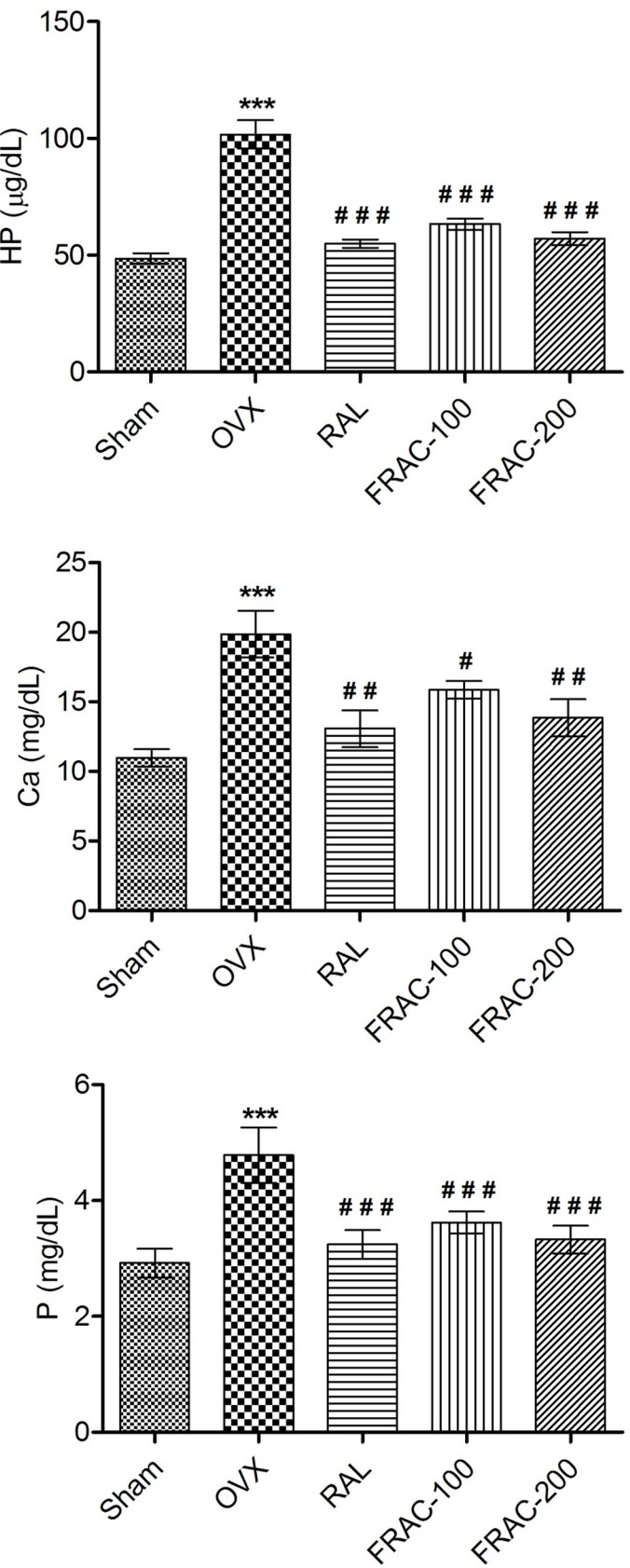

**Fig 2. Effect of FRAC from *Pueraria tuberosa* on biochemical parameters of urine.** Data were average ± SEM (n = 6). *** p < 0.01 significantly different from sham control group. # p < 0.05, ## p < 0.01, ### p < 0.001 significantly different from OVX group. Ca, calcium; P, phosphorus; HP, hydroxyproline.

**Effect of FRAC on biomechanical parameters.** None of the groups showed any significant alteration of femur length. Femur thickness, volume, weight, and breaking strength were significantly decreased (p<0.001) in the OVX control compared to the sham control group. A significant increase (p<0.05) in all these parameters (Figs 3 and 4) was observed with FRAC and raloxifene administration. Furthermore, OVX caused a significant reduction (p<0.01) of femur density, and treatment with FRAC showed a substantial improvement (p<0.05) of the femur density. Treatment with raloxifene and FRAC restored the breaking strength of the 4th lumbar vertebrae caused by ovariectomy (Fig 4).

**Effect of FRAC on body and organ weight.** A significant (p<0.001) increase in body weight (BW) was observed due to OVX though there was no significant variation at the start of the study. Treatment with FRAC and raloxifene markedly reduced (p<0.01) the increased BW (Fig 5) as well as the final and initial BW difference, compared to OVX. The OVX caused a marked reduction (p<0.001) in the uterus weight. In comparison, the administration of raloxifene, and FRAC significantly increased (p<0.001) uterine weight compared to the OVX group

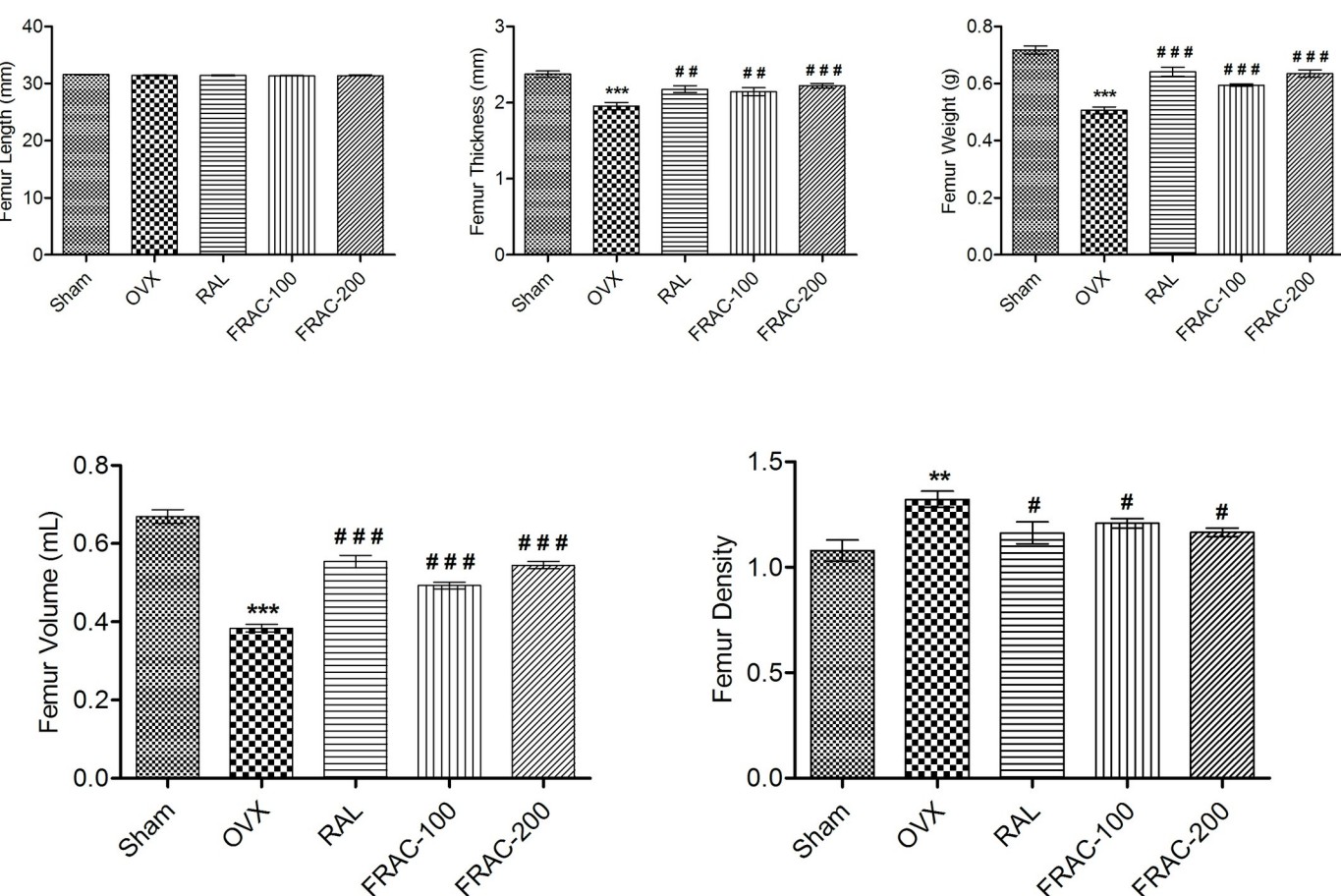

**Fig 3. Effect of FRAC from *Pueraria tuberosa* on femur biomechanical parameters.** Data were average ± SEM (n = 6). ** p < 0.01, *** p < 0.001 significantly different from sham control group. # p < 0.05, ## p < 0.01, ### p < 0.001 significantly different from OVX group.

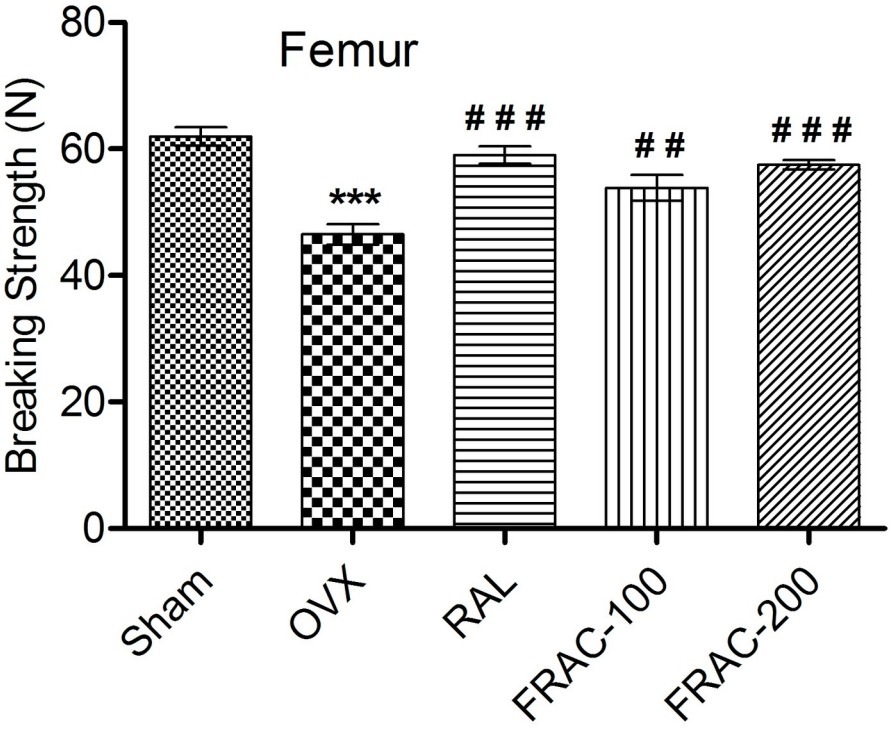

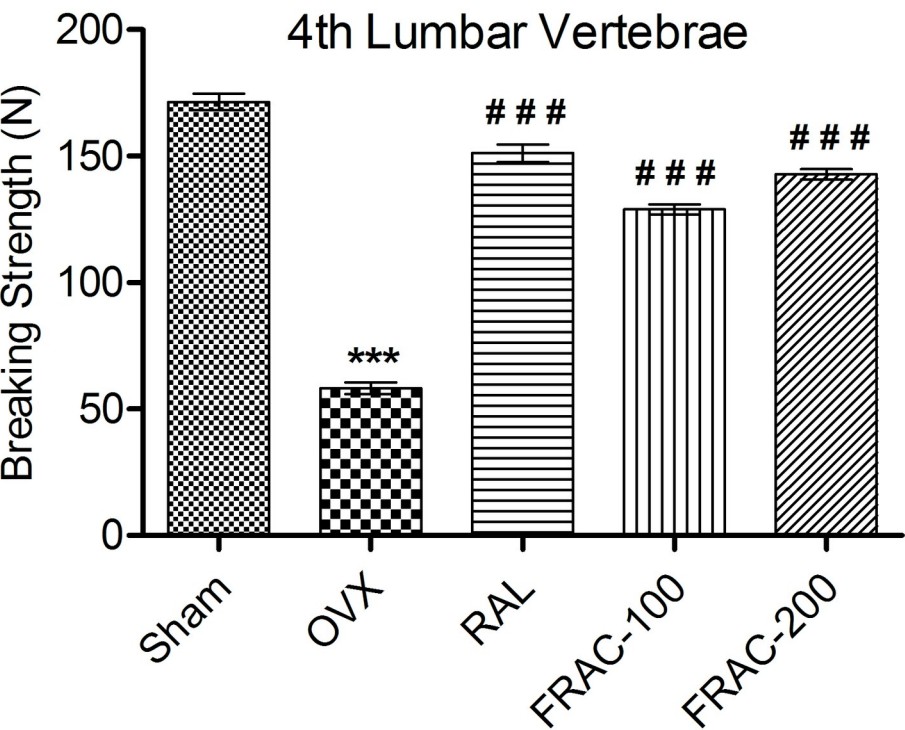

**Fig 4. Effect of FRAC from *Pueraria tuberosa* on breaking strength of femur and 4ᵗʰ lumbar vertebrae.** Data were average ± SEM (n = 6). *** $p < 0.001$ significantly different from sham control group. ## $p < 0.01$, ### $p < 0.001$ significantly different from OVX group.

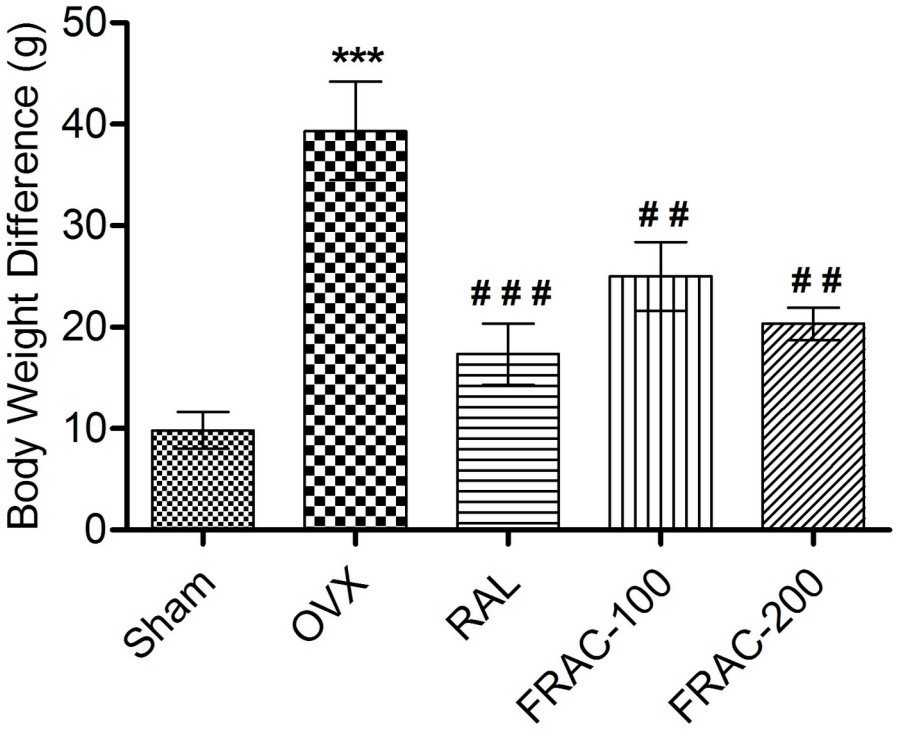

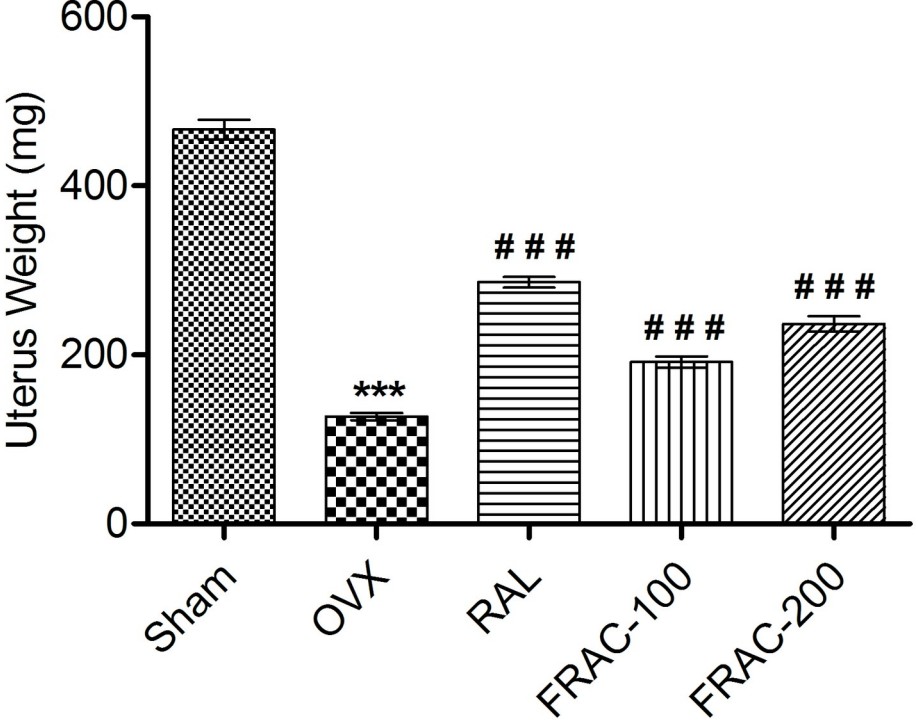

**Fig 5. Effect of FRAC from *Pueraria tuberosa* on body and uterus weight.** Data were average ± SEM (n = 6). ***
$p < 0.001$ significantly different from Sham control group. ## $p < 0.01$, ### $p < 0.001$ significantly different from OVX
group.

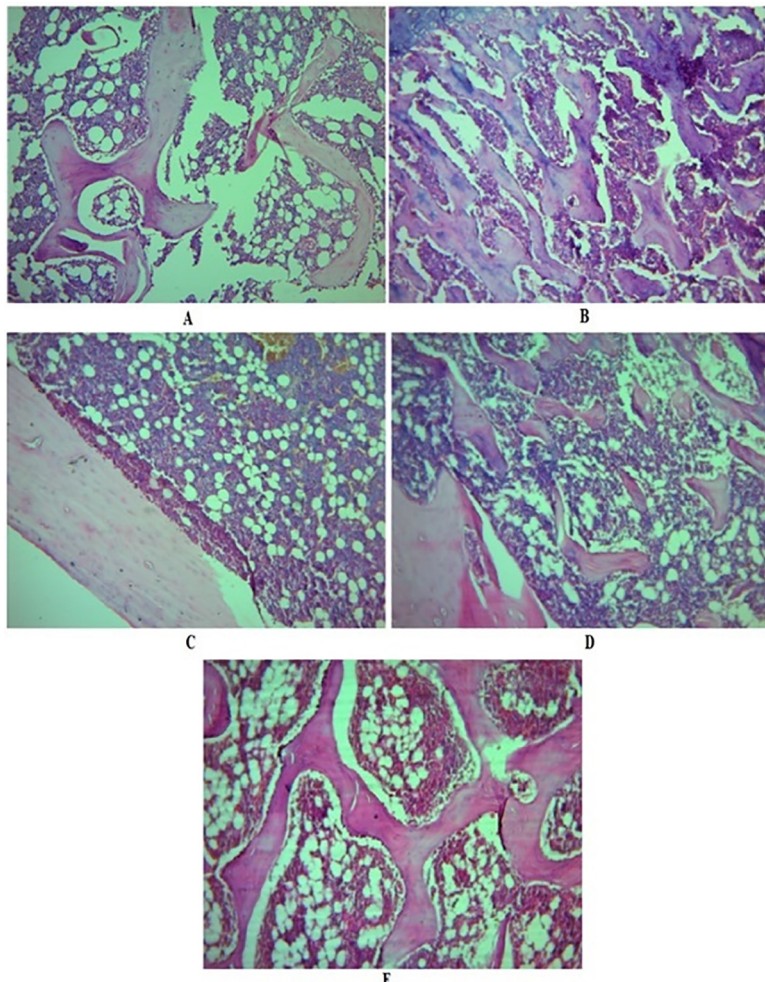

**Fig 6. Effect of FRAC of *Pueraria tuberosa* on the histopathology of the femur. A,** Photomicrography of the femur of sham control group showing typical bone architecture; **B,** Photomicrography of the femur of OVX control group showing disruption of trabeculae; **C,** Photomicrography of the femur of raloxifene treated group showing improved trabecular thickness, and compactness of cells indicating mineralization of bone; **D,** Photomicrography of the femur of FRAC-100 mg/kg treated group showing the improved trabecular thickness and bone architecture; **E,** Photomicrography of the femur of FRAC-200 mg/kg treated group showing the restoration of typical bone architecture and an increase in width of trabeculae.

(Fig 5). The percentage of uterus weight relative to BW for group I, II, III, IV and V animals was 0.19, 0.05, 0.11, 0.07, and 0.09, respectively.

**Histopathology study.** Photomicrographs of the femur of different groups of animals are depicted in Fig 6A–6E. There was a distraction of trabeculae with the decline in thickness and development of large cyst-like spaces following OVX. The photographs show trabecular ossification, mineralization, and compactness in the groups treated with raloxifene and FRAC. Raloxifene and FRAC generated antiosteoporotic activity in OVX rats.

## *In vitro* cytotoxicity of FRAC

Postmenopausal osteoporosis, which typically causes weakness of bone as the process of bone-resorption exceeds bone-formation because of the estrogen-deficient state [34]. Hormone replacement therapy (HRT) is a choice to manage the problems in postmenopausal women,

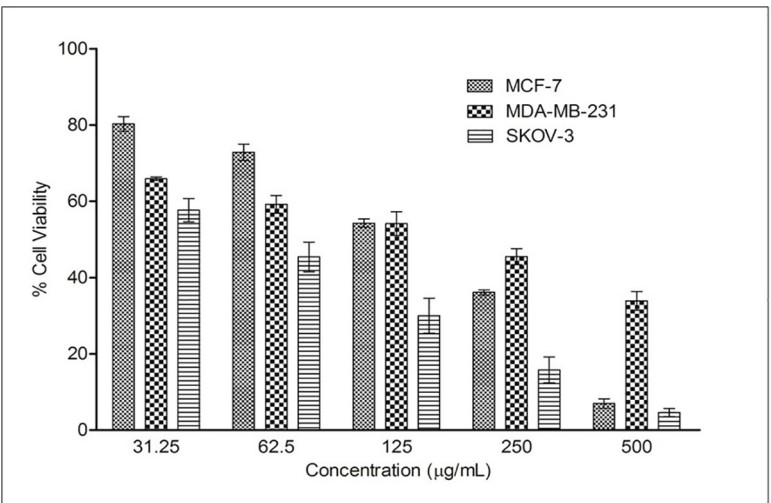

**Fig 7. *In vitro* cytotoxicity of FRAC of *Pueraria tuberosa* against different cancer cell lines.** Values are mean ± SEMs (n = 3).

but continuous administration of HRT has the danger of cancer (breast, ovary, and endometrial) development [35]. Therefore, we assessed the *in vitro* anticancer activity of FRAC (31.5–500 μg/mL) in breast (MCF-7 and MDA-MB-231) and ovarian (SKOV-3) cancer cells. FRAC displayed anticancer activity against the three cancer cell lines in a dose-dependent manner (Fig 7). FRAC demonstrated better activity against ovarian cancer cells compared to breast cancer cells.

## Docking study

High-performance thin layer chromatography analysis confirmed the presence of daidzein and genistein in FRAC of *Pueraria tuberosa* [31]. Docking pose of these phytoconstituents into estrogen receptor (ER) α (1 X 76) and β (1 X 7R) were evaluated and furnished in Figs 8

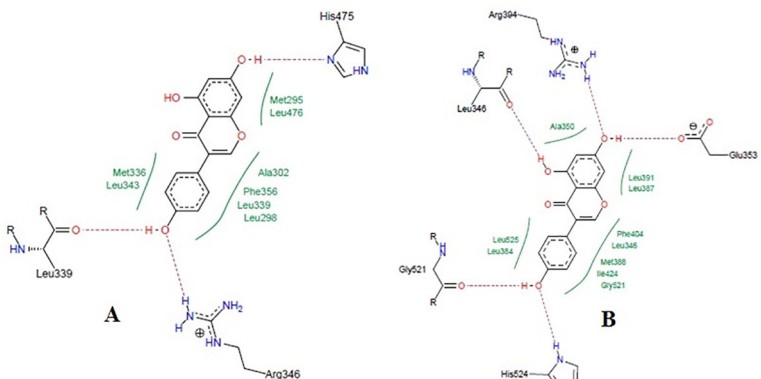

**Fig 8. Docking study of genistein in estrogen receptors. A,** Co-crystalized ligand of 1x76 (genistein) showing hydrogen bond with Arg346, Leu339, and His475. Hydrophobic interactions were also seen near the benzene rings with different amino acid residues of estrogen receptor α (PDB: 1x76). The ligand showed a docking score of -26.1648. **B,** Co-crystalized ligand of 1x7R (genistein) showing hydrogen bond with Leu346, Arg394, Gly521, Glu353, and His524. Hydrophobic interactions were also seen near the benzene rings with different amino acid residues of estrogen receptor β (PDB: 1x7R). The ligand showed a docking score of -32.4084.

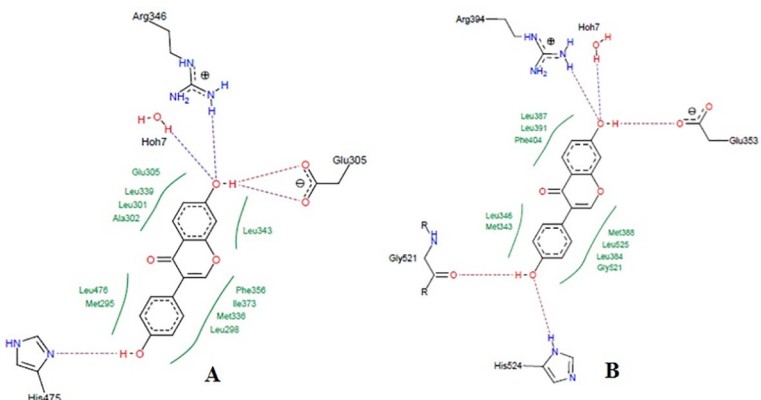

**Fig 9. Docking study of daidzein in estrogen receptors. A,** Docking pose of daidzein in estrogen receptor α (PDB: 1x76) active site with a docking score of -28.3129. Daidzein formed a hydrogen bond with Arg346, Glu305, and His475. Hydrophobic interactions were also seen near the benzene rings with different amino acid residues of estrogen receptor α (PDB: 1x76). **B,** Docking pose of daidzein in estrogen receptor β (1 x 7R) active site with a docking score of -31.8923. Daidzein showed a hydrogen bond with Arg394, Glu353, Gly521 and His524. Hydrophobic interactions were also seen near the benzene rings with different amino acid residues of estrogen receptor β (PDB: 1x7R).

and 9. Genistein exhibited -26.1648 and -32.4084 docking score into ER- α and β active site, respectively. The docking score of daidzein into ER- α and β active site were -28.3129 and -31.8923, respectively. However, ER expression could be carried out in future. Docking analysis of internal ligands is shown in Fig 10.

The ADMET profiles for these compounds as well as the drug-likeness prediction of the compounds are provided in Tables 3 and 4. Drug-likeness refers to the possibility of a molecule to become an oral drug regarding its bioavailability. Five different filters were employed to determine the drug & lead likeness for daidzein and genistein (Table 3). Both compounds revealed good drug-likeness score with zero violation of drug-likeness rules, and also exhibited a lead-likeness with no violation. They did not show any violations with reference to PAINS

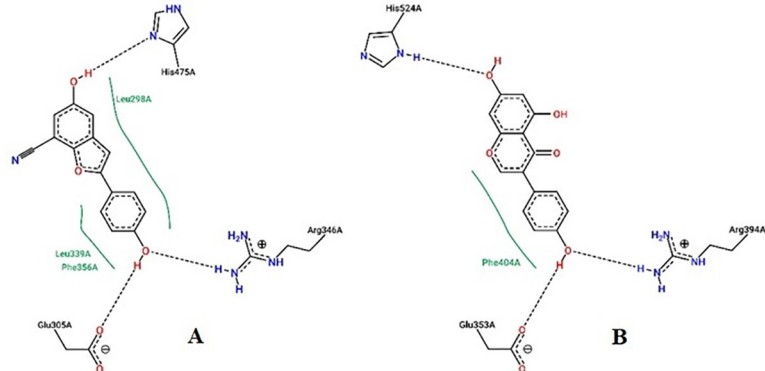

**Fig 10. Docking study of internal ligands on Ix76 and 1x7R receptors. A,** Docking pose of daidzein in estrogen receptor α (PDB: 1x76) active site with a docking score of -28.3129. Daidzein formed hydrogen bond with Arg346, Glu305, and His475. Hydrophobic interactions were also seen near the benzene rings with different amino acid residues of estrogen receptor α (PDB: 1x76). **B,** Docking pose of 5-hydroxy-2-(4-hydroxyphenyl)-1-benzofuran-7-carbonitrile in estrogen receptor β (1 x 7R) active site with a docking score of -28.9356. 5-hydroxy-2-(4-hydroxyphenyl)-1-benzofuran-7-carbonitrile showed a hydrogen bond with Arg394, Glu353and His524. Hydrophobic interactions were also seen near the benzene rings with different amino acid residues of estrogen receptor β (PDB: 1x7R).

**Table 3. Details of different drug-likeness rules, bioavailability, lead-likeness, synthetic accessibility, and alerts for PAINS and Brenk.**

| Compound | Drug-likeness Rules | | | | | | Alerts | | Lead likeness | Synthetic Accessibility |
|---|---|---|---|---|---|---|---|---|---|---|
| | Lipinski (Pfizer) | Ghose (Amgen) | Veber (GSK) | Egan (Pharmacia) | Muege (Bayer) | Bioavailability Score | PAINS | Brenk | | |
| Daidzein | Yes | Yes | Yes | Yes | Yes | 0.55 | 0 | 0 | Yes | 2.79 |
| Genistein | Yes | Yes | Yes | Yes | Yes | 0.55 | 0 | 0 | Yes | 2.87 |

and Brenk method to determine the probable uncertain fragments that yield false-positive biological output. Lead likeness was also estimated along with the synthetic accessibility appraisal. The compounds that showed high scores were removed as they were tough to synthesize as per protocols. Daidzein and genistein could be synthesized easily as their scores were 2.79 and 2.87, respectively. Further, both the compounds have a bioavailability score of 55% (Table 3) indicating good oral absorption.

## Discussion

Antioxidants play a major role in controlling menopausal complications, including osteoporosis [7]. In this study, we explored the *in vivo* antiosteoporotic and *in vitro* anticancer activities of an FRAC from the tubers of *Pueraria tuberosa*. Ethanol extract of tubers of *Pueraria tuberosa* and its various fractions (hexane, ethyl acetate, n-butanol, and aqueous) were analyzed for total phenolic and flavonoid content, and antioxidant activity. The ethyl acetate fraction showed maximum phenolic and flavonoid content, and antioxidant property (Tables 1 and 2), and was selected as the FRAC.

The antiosteoporotic activity of the FRAC was evaluated in OVX rats by determining biochemical and biomechanical parameters, body and organ weights, and histopathology (Figs 1–6). The pattern of change in bone mineral parameters such as P and Ca in the present study confirms earlier findings of minor bone mineralization and balanced mineral homeostasis [4]. The FRAC did not change homeostasis, which may be due to enhanced absorption of calcium

**Table 4. Details of in-silico ADME profile of flavonoids using Swiss ADME online server.**

| ADMET PROFILE | | | Daidzein | Genistein |
|---|---|---|---|---|
| | Physiochemical parameters | Formula | $C_{15}H_{10}O_4$ | $C_{15}H_{10}O_5$ |
| | | Molecular weight | 254.24 g/mol | 270.24 g/mol |
| | | Mol. refractivity | 71.97 | 73.99 |
| | | TPSA | 70.67 Å$^2$ | 90.90 Å$^2$ |
| | Lipophilicity | ILOGP | 1.77 | 1.91 |
| | | SILICOS-IT | 3.02 | 2.52 |
| | Water Solubility | Log S (ESOL), Class | -3.53 | -3.72 |
| | | Log S (Ali), Class | -3.60 | -4.23 |
| | | SILICOS-IT, Class | -4.98 | -4.40 |
| | Pharmacokinetics | GI absorption | High | High |
| | | BBB permeant | Yes | No |
| | | Log $K_p$ (skin perm.) | -6.10 cm/s | -6.05 cm/s |
| | | CYP1A2 | Yes | Yes |
| | | CYP2C19 | No | No |
| | | CYP2C9 | No | No |
| | | CYP2D6 | Yes | Yes |
| | | CYP3A4 | Yes | Yes |

from the intestine [4, 7]. ALP and TRAP (bone turnover markers) activity are signs of bone osteoblast functioning and factors of bone formation. OVX increased the level of these markers in serum because of the reduction in the estrogen level. HP is commonly accepted as a biochemical parameter associated with bone metabolism, and its level is a sign of osteogenic activity. Urinary HP which indicates a breakdown of collagen due to the high level of TRAP is formed from activated osteoclast [36]. Increased level of HP, TRAP, and ALP was observed in OVX group. Higher levels of these biochemical parameters indicate decreased bone formation and augmentation of collagen degradation. Administration of raloxifene and FRAC reversed this effect of the OVX indicating the bone resorption inhibition property of raloxifene and FRAC. FRAC treatment produced positive effects on OVX-induced hyperlipidemia which could be due to the existence of daidzein, genistein, and β-sitosterol in *Pueraria tuberosa*. Flavonoids are scavengers of reactive oxygen species (ROS). The ROS block TG secretion into the plasma and upset cholesterol catabolism into bile acids. Daidzein and genistein possess antihyperlipidemic effects [37]. Further, the presence of β-sitosterol in FRAC hinders the absorption of cholesterol by controlling lipogenesis and lipolysis [38].

It has been shown that reduction in bone mineral density in patients leads to bone loss and increased susceptibility of fracture [39]. Healthy bones are normally compact and can tolerate considerable load. The compactness of the bone could be assessed by determining bone strength. Ovariectomized (OVX) animal model, used in the current study is a common screening method for antiosteoporotic agents. The breaking strength of femur and 4th lumbar vertebrae was substantially increased by FRAC of *Pueraria tuberosa* showing the protective effect of FRAC against menopausal osteoporosis. The antiosteoporotic effect of the FRAC of *Pueraria tuberosa* is comparable to earlier studies with other plant materials [4, 7]. The phytoestrogens of FRAC might have augmented the osteoclast activity and reduced bone turnover.

The reduction of estrogen level in OVX animals causes an increase in energy intake and elevated body weight [40]. Further, the decrease in estrogen level due to OVX led to the deposition of fat (as shown in the rise of total cholesterol and triglyceride in serum) and hence an increase in body weight [11]. The FRAC administration reduced the level of cholesterol and TG in serum, and decreases the OVX-induced gain in body weight. These observations support the protective effect of FRAC against fat deposition and weight gain in menopausal osteoporosis [41].

Bone strength is associated with bone mass, as well as its structure. Hence, histopathological analysis is a significant parameter to analyze bone strength. OVX is associated with an increase in bone turnover, reduction in bone balance and loss in bone mineral density in the trabecular region of the femur [42]. The osteoprotective property of FRAC of *Pueraria tuberosa* manifested by superior trabecular architecture may be attributed its secondary metabolites that may work as phytoestrogens and minimize bone loss [43].

In our previous study, we reported the presence of two isoflavones, genistein and daidzein in the FRAC of *Pueraria tuberosa* [31]. In the present study, the phytoestrogenic nature of these two isoflavones was established by docking studies with estrogen receptor α and β, where both the compounds were found to have a good affinity with both the receptors. Estradiol has docking scores of -18 and -17 into estrogen receptor α and β active site, respectively [41]. The bioactive compounds, genistein and daidzein present in *Pueraria tuberosa* have higher affinity compared to estradiol, which is also supported by earlier studies as they showed high affinity for estrogen receptors [44–46]. These two compounds may be mainly responsible for the antiosteoporotic property of the FRAC of *Pueraria tuberosa*.

Phytoestrogens have been used as an alternative therapy for the management of menopausal osteoporosis as the regular use of hormone replacement therapy causes severe side effects, including cancer of the breast and ovary [35]. Phytoestrogenic compounds also induce

cell proliferation in ER-positive human breast cancer cells (MCF-7) [47]. However, in this study, FRAC exhibited anticancer property in breast and ovarian cancer cells, suggesting its suitability for the treatment of postmenopausal osteoporosis.

## Conclusion

The FRAC of *Pueraria tuberosa* exhibited marked antiosteoporotic activity in ovariectomy-induced osteoporosis. Part of the protective effect of FRAC might be attributed to its antioxidant potential as bone loss in osteoporosis could be due to the generation of reactive oxygen species/oxidative stress along with other factors. Also, the presence of phytoestrogenic compounds such as daidzein and genistein in the FRAC and their binding to estrogen receptors may add to the protective effect of the FRAC. The FRAC also exhibited significant anticancer activity in breast and ovarian cancer cells. The findings indicate that FRAC has great potential as therapeutics for controlling menopausal osteoporosis with the additional benefit of anticancer effect. However, further mechanistic studies including the redox status in bone, HDL & LDL levels in serum, and safety studies are needed for potential use of the FRAC of *Pueraria tuberosa* in the therapy of osteoporosis.

## Supporting information

**S1 Table. Chemical compounds in FRAC from *Pueraria tuberosa* by GC/MS analysis.**
(DOC)

## Acknowledgments

The support of Dr. Pankaj Samuel, GC/MS Laboratory, Panjab University, Chandigarh, India, is highly appreciated for performing GC/MS analysis. We would like to express our sincere gratitude to Dr. Manik Ghosh, Department of Pharmaceutical Sciences and Technology, Birla Institute of Technology, Ranchi, India, for conducting docking studies.

## Author Contributions

**Conceptualization:** Swaha Satpathy, Arjun Patra, Bharti Ahirwar.

**Data curation:** Swaha Satpathy, Arjun Patra.

**Formal analysis:** Swaha Satpathy, Arjun Patra, Muhammad Delwar Hussain.

**Funding acquisition:** Arjun Patra.

**Investigation:** Swaha Satpathy, Arjun Patra.

**Methodology:** Swaha Satpathy, Arjun Patra, Muhammad Delwar Hussain.

**Project administration:** Bharti Ahirwar.

**Software:** Swaha Satpathy, Arjun Patra.

**Supervision:** Bharti Ahirwar.

**Validation:** Muhammad Delwar Hussain, Mohsin Kazi.

**Writing – original draft:** Swaha Satpathy.

**Writing – review & editing:** Swaha Satpathy, Arjun Patra, Muhammad Delwar Hussain, Mohsin Kazi, Mohammed S. Aldughaim, Bharti Ahirwar.

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
