## [Decision Letter · Decision Letter 0]

14 Oct 2020

PONE-D-20-26098

Antioxidant enriched fraction from Pueraria tuberosa alleviates ovariectomized-induced osteoporosis in rats, and inhibits growth of breast and ovarian cancer cell lines in vitro

PLOS ONE

Dear Dr. Satpathy,

Thank you for submitting your manuscript to PLOS ONE. After careful consideration, we feel that it has merit but does not fully meet PLOS ONE’s publication criteria as it currently stands. Therefore, we invite you to submit a revised version of the manuscript that addresses the points raised during the review process.

Reviewer #1 found the manuscript deficient in several values which should be determined in additional measurements/experiments in order to provide a solid scientific backgound for the recorded observations and concluded claims. Statistical testing should be presented in a more profound manner, while the language is to be considerably polished throughout the text. Please, be aware of the attachment with additional comments provided by Reviewer #2.

We look forward to receiving your revised manuscript.

Kind regards,

Branislav T. Šiler, Ph.D.

Academic Editor

PLOS ONE

Journal Requirements:

2) PLOS requires an ORCID iD for the corresponding author in Editorial Manager on papers submitted after December 6th, 2016. Please ensure that you have an ORCID iD and that it is validated in Editorial Manager. To do this, go to ‘Update my Information’ (in the upper left-hand corner of the main menu), and click on the Fetch/Validate link next to the ORCID field. This will take you to the ORCID site and allow you to create a new iD or authenticate a pre-existing iD in Editorial Manager. Please see the following video for instructions on linking an ORCID iD to your Editorial Manager account: https://www.youtube.com/watch?v=_xcclfuvtxQ

3) Please include a copy of Table 3 which you refer to in your text on page 24.

4) Please include captions for your Supporting Information files at the end of your manuscript, and update any in-text citations to match accordingly. Please see our Supporting Information guidelines for more information: http://journals.plos.org/plosone/s/supporting-information.

5) We noticed there is a bit of text overlap with this previously published paper of yours: https://doi.org/10.1016/j.sajb.2018.05.033, which will need to be addressed. In your revision ensure you cite all your sources (including your own work), and quote or rephrase any duplicated text outside the methods section. Further consideration is dependent on these concerns being addressed.

Reviewers' comments:

Reviewer's Responses to Questions

**Comments to the Author**

1. Is the manuscript technically sound, and do the data support the conclusions?

Reviewer #1: Yes

Reviewer #2: Partly

2. Has the statistical analysis been performed appropriately and rigorously? 

Reviewer #1: Yes

Reviewer #2: Yes

3. Have the authors made all data underlying the findings in their manuscript fully available?

Reviewer #1: Yes

Reviewer #2: Yes

4. Is the manuscript presented in an intelligible fashion and written in standard English?

Reviewer #1: No

Reviewer #2: No

5. Review Comments to the Author

Reviewer #1: The manuscript is a good research area. However, there are few grammatical errors that must be check in the manuscript. All comments raised have been uploaded for authors. The basis for the dosages used in the methodology must be provided by authors.

Reviewer #2: The present submission claimed that AEF may alleviate bone loss through its anti-oxidant activity. However, I have a few concerns about the submission.

(1) Docking study demonstrated AEF may bind to ERa and ERb. Therefore, ER expression should be examined in the bone.

(2) The uterus weight relative to body weight should be determined.

(3) The authors claimed that AEF may inhibit hyperlipidemia. However, the histological staings in the provided figure appeared that there is less fat disposition in the femur. In addition, oil red O staings should be evaluated in the bone. Moreover, what is the alterations of serum HDL and LDL?

(4) The redox status should be tested in the bone.

(5) If possible, the BMD and bone microstructure should be evaluated.

(6) The P value should be followed when the authors mentioned the result is significant.

(7) Please provide the batch number for the kits and chemicals.

(8) Grammatical mistakes have been found in the MS.

(9) Please eliminate the description of the anticancer. The current data did not support the assertion.

6. PLOS authors have the option to publish the peer review history of their article (what does this mean?). If published, this will include your full peer review and any attached files.

Reviewer #1: No

Reviewer #2: No

---

## [Author Response · Author response to Decision Letter 0]

9 Dec 2020

Below we present our point by point responses to the reviewer/editorial comments:

Reviewer/Editorial Comments Response

"Main title: "Antioxidant enriched fraction" has no meaning. Instead "Antioxidant" it can stand "Antioxidant compounds". Moreover,

 "enriched" sounds like the authors somehow added antioxidant compounds to the fraction they have investigated. Instead of "enriched",

"rich"(if they are clearly reach) or "containing" should read (throughout the text). And finally, the authors did not investigate a fraction of P. tuberosa, but a fraction of its extract." We have corrected "Antioxidant enriched fraction" to "Antioxidant-rich fraction" throughout the manuscript.

The title is changed as: Antioxidant-rich fraction from Pueraria tuberosa alleviates ovariectomized-induced osteoporosis in rats, and inhibits growth of breast and ovarian cancer cells

1) Please ensure that your manuscript meets PLOS ONE's style requirements, including those for file naming. The PLOS ONE style templates can be found at Corrected as per the PLOS ONE style templates.

2) PLOS requires an ORCID iD for the corresponding author in Editorial Manager on papers submitted after December 6th, 2016. Please ensure that you have an ORCID iD and that it is validated in Editorial Manager. To do this, go to ‘Update my Information’ (in the upper left-hand corner of the main menu), and click on the Fetch/Validate link next to the ORCID field. This will take you to the ORCID site and allow you to create a new iD or authenticate a pre-existing iD in Editorial Manager. ORCID iD information is added.

3) Please include a copy of Table 3 which you refer to in your text on page 24. Table 3 was mentioned in the MS by mistake and now it has been corrected as the supplementary material S1.

4) Please include captions for your Supporting Information files at the end of your manuscript, and update any in-text citations to match accordingly. Captions are included and updated the in-text citations

5) We noticed there is a bit of text overlap with this previously published paper of yours: https://doi.org/10.1016/j.sajb.2018.05.033, which will need to be addressed. In your revision ensure you cite all your sources (including your own work), and quote or rephrase any duplicated text outside the methods section. Further consideration is dependent on these concerns being addressed. We have extensively edited, rephrased the revised manuscript for any text overlap with the previous paper.

Reviewer's Responses to Questions 

1. Is the manuscript technically sound, and do the data support the conclusions?

The manuscript must describe a technically sound piece of scientific research with data that supports the conclusions. Experiments must have been conducted rigorously, with appropriate controls, replication, and sample sizes. The conclusions must be drawn appropriately based on the data presented. Reviewer #1: Yes

Reviewer #2: Partly

Changes and corrections have been made in the manuscript to address the comment for Reviewer #2

2. Has the statistical analysis been performed appropriately and rigorously? Reviewer #1: Yes

Reviewer #2: Yes

3. Have the authors made all data underlying the findings in their manuscript fully available? Reviewer #1: Yes

Reviewer #2: Yes

4. Is the manuscript presented in an intelligible fashion and written in standard English?

PLOS ONE does not copyedit accepted manuscripts, so the language in submitted articles must be clear, correct, and unambiguous. Any typographical or grammatical errors should be corrected at revision, so please note any specific errors here. Reviewer #1: No

Reviewer #2: No

The manuscript is extensively edited to improve the language. The typographical or grammatical errors have also been corrected.

5. Review Comments to the Author 

Reviewer #1: The manuscript is a good research area. However, there are few grammatical errors that must be check in the manuscript. All comments raised have been uploaded for authors. The basis for the dosages used in the methodology must be provided by authors. We thank the reviewer for making inspiring comments. We have gone through the manuscript very carefully and corrected grammatical, and spelling mistakes as per the suggestions. The basis for the dose use is provided below.

1) Why and what is the basis for this dose selection? No mortality or any signs of moribund status were found at this dose (2000 mg/kg). Therefore, the LD50 cut-off is 5000 mg/kg (category 5 in the Globally Harmonized Classification System). Usually, 1/5th and 1/10th of the LD50 value can be used for animal experimentation and the dose selected (very small dose compared to LD50 cut-off, 5000 mg/kg) for screening antiosteoporotic activity, where the treatment was for 90 days. We have not observed and mortality of signs of toxicity during this 90 days period also. This information is now provided in the manuscript (page #, lines #-#)

2) In the molecular docking methodology, it was necessary to detail more, for example, which values of the grid box (x, y and z). 1x7R: 15.587, 32.224, 22.304

1x76: 29.554, 37.747, 38.951

This information is now provided in the manuscript (page 14, lines 284-285)

No proper methodology in ligand preparation. Provide detailed methodology. 3D conformer (.sdf) of daidzein and genistein were downloaded from PubChem website and used as ligand. FlexXLeadIT 2.1.8 of BiosolveIT uses only 3D conformer of ligands. This information is now provided in the manuscript (page 14, lines 292-293)

Genistein: https://pubchem.ncbi.nlm.nih.gov/compound/5280961

Daidzein: 

https://pubchem.ncbi.nlm.nih.gov/compound/5281708

The information is added in the method section (page#, lines #-#)

Why was a reference drug not used to compare with the two compounds used for docking studies? 5-hydroxy-2-(4-hydroxyphenyl)-1-benzofuran-7-carbonitrile was used as internal ligand in 1X76 (ER-α). Genistein was the internal ligand of 1X7R (ER-β), Figure 10 mentioned in page 22-23. Further, As per earlier literature, estradiol (a reference drug) has docking score of -18 and -17 into estrogen receptor α and β active site, respectively, which is mentioned in the discussion section (Page 27, line 548-549).

Also, authors should provide the ADMET profile for these compounds as well as the Drug likeness prediction The ADMET profile of the two compounds are now provided in the manuscript (materials section page 14-15, and results section page 23-24)

Reviewer #2: The present submission claimed that AEF may alleviate bone loss through its anti-oxidant activity. However, I have a few concerns about the submission. 

(1) Docking study demonstrated AEF may bind to ERa and ERb. Therefore, ER expression should be examined in the bone. The docking study was done to support the data obtained in animal study and we found good interaction of daidzein and genistein with both ER-α and ER-β. However, ER expression could be carried out in future. We have mentioned it in the revised manuscript (page 22, line 449).

(2) The uterus weight relative to body weight should be determined. The weight ratios are: Gr-I (0.1896%); Gr-II (0.0457%); Gr-III (0.1124%); Gr-IV (0.0734%); Gr-V (0.0918%). We have now included this numbers in the revised manuscript (page 20, line 405-407)

(3) The authors claimed that AEF may inhibit hyperlipidemia. However, the histological staings in the provided figure appeared that there is less fat disposition in the femur. In addition, oil red O staings should be evaluated in the bone. Moreover, what is the alterations of serum HDL and LDL? We thank the reviewer for pointing this out. We made the statement based on the ability of AEF to reduce the OVX-induced increased level of triglycerides and total cholesterol. We have not measured the serum HDL and LDL levels at the time of animal experiment.

(4) The redox status should be tested in the bone. Thanks for the suggestion. As we have not measured the balance between oxidants and antioxidants during the experiment. 

(5) If possible, the BMD and bone microstructure should be evaluated. We thank the reviewer for the great suggestion. But we could not evaluate the BMD and bone microstructure due to lack of facility at the time of the animal experiment.

(6) The P value should be followed when the authors mentioned the result is significant. We thank the reviewer for the excellent suggestion. We have now provided the information in the manuscript (results section in pages 17-20).

(7) Please provide the batch number for the kits and chemicals. Thanks to the reviewer for the suggestion. We have now provided the information here, but not mentioned in the MS. Currently, we do not have the batch number of some of the reagents/chemicals as they were over and not provided here.

3-(4,5-dimethylthiazol-2-yl)-2,5-diphenyltetrazoliumbromide (MTT): lot #11206BE

Dimethyl sulfoxide (DMSO): Lot # S0154

Phosphate Buffered Saline (PBS): Lot # 11419005

Xylazine: FFK8002

Ketamine: 382053

Gentamicin: AHF0051

Calcium kit: CAO-020

Phosphorus kit: PHO-L-B1002

Urea kit: URE-L-B1010

Alkaline phosphatase kit: ALP-L-B1086

Cholesterol kit: HDL-L-B1046

Triglyceride kit: TGR-L-B1106

(8) Grammatical mistakes have been found in the MS. We have searched and corrected the grammatical mistakes Throughout the manuscript

(9) Please eliminate the description of the anticancer. The current data did not support the assertion. The Antioxidant-Rich Fraction (ARF) has the potential for treatment of menopausal osteoporosis. In addition, it has anticancer effect in cancer cells. This is advantageous as treatment with estrogens have the side effect including possibility of developing cancer.To make this clear,we have mentioned this in the Introduction section (page 4, paragraph 2), discussion (page 27, lines 554-559), conclusion (page 30, lines 567-568), and abstract (page 2, lines 44)

Dr. Swaha Satpathy

---

## [Decision Letter · Decision Letter 1]

11 Dec 2020

PONE-D-20-26098R1

Antioxidant-rich fraction from Pueraria tuberosa alleviates ovariectomized-induced osteoporosis in rats, and inhibits growth of breast and ovarian cancer cells

PLOS ONE

Dear Dr. Satpathy,

Thank you for submitting your manuscript to PLOS ONE. After careful consideration, we feel that it has merit but does not fully meet PLOS ONE’s publication criteria as it currently stands. Therefore, we invite you to submit a revised version of the manuscript that addresses the points raised during the review process.

The authors did not properly respond to the comments provided by the Reviewer #2. Two major issues are stated in the reviewers' reports below.

In addition, please take notice of the following comments:

The authors did not grasp the Editor's comments related to the proper usage of scientific terminology introduced in the previous review round. Let me try to explain in a semantic manner:

"Antioxidant" is an adjective and cannot present an object. Therefore, a subject cannot be rich in an attribute, but in some item (noun). The expression "Antioxidant compounds" sound like a logical choice as suggested in the previous review round.

The phrase "fraction from *Pueraria tuberosa*" is unaccomplished, vague. Can authors define how would they describe a fraction of a plant? Instead of that, a fraction of **a plant extract** would be more appropriate.

Considering above mentioned, I would suggest transforming the main title into: "A fraction of *Pueraria tuberosa* extract, rich in antioxidant compounds, alleviates ovariectomized-induced osteoporosis in rats and inhibits growth of breast and ovarian cancer cells". The same terminology has to be applied throughout the text.

L23: No need to introduce the abbreviation of the genus name, i.e. please delete the bracket. Instead of that, the species authority should be introduced as: *Pueraria tuberosa* (Roxb. ex Willd.) DC. here and also in the L90. In L89: all the Latin names must stand in italics as well as elsewhere in the text.

Compound names are randomly capitalized. Please do not write compound names with capital letters unless they present trademarks.

ARF (antioxidant-rich fraction) cannot stand from the reasons stated as above. "Fraction rich in antioxidant compounds" and a matching abbreviation can read.

Figures 8 and 9 are vertically stretched out. Can you revert them to the original ratio?

We look forward to receiving your revised manuscript.

Kind regards,

Branislav T. Šiler, Ph.D.

Academic Editor

PLOS ONE

Reviewers' comments:

Reviewer's Responses to Questions

**Comments to the Author**

1. If the authors have adequately addressed your comments raised in a previous round of review and you feel that this manuscript is now acceptable for publication, you may indicate that here to bypass the “Comments to the Author” section, enter your conflict of interest statement in the “Confidential to Editor” section, and submit your "Accept" recommendation.

Reviewer #1: All comments have been addressed

Reviewer #2: (No Response)

2. Is the manuscript technically sound, and do the data support the conclusions?

Reviewer #1: Yes

Reviewer #2: Yes

3. Has the statistical analysis been performed appropriately and rigorously? 

Reviewer #1: Yes

Reviewer #2: Yes

4. Have the authors made all data underlying the findings in their manuscript fully available?

Reviewer #1: Yes

Reviewer #2: Yes

5. Is the manuscript presented in an intelligible fashion and written in standard English?

Reviewer #1: Yes

Reviewer #2: Yes

6. Review Comments to the Author

Reviewer #1: All comments have been addressed. Research ethics was followed as ethical approval was obtained for this study.

Reviewer #2: The authors partly responsed my comments. However,

(1) The redox status should be tested in the bone.

(2) The author should explain why did not test HDL and LDL.

7. PLOS authors have the option to publish the peer review history of their article (what does this mean?). If published, this will include your full peer review and any attached files.

Reviewer #1: No

Reviewer #2: No

---

## [Author Response · Author response to Decision Letter 1]

22 Dec 2020

1. "Antioxidant" is an adjective and cannot present an object. Therefore, a subject cannot be rich in an attribute, but in some item (noun). The expression "Antioxidant compounds" sound like a logical choice as suggested in the previous review round.

The phrase "fraction from Pueraria tuberosa" is unaccomplished, vague. Can authors define how would they describe a fraction of a plant? Instead of that, a fraction of a plant extract would be more appropriate.

Considering above mentioned, I would suggest transforming the main title into: "A fraction of Pueraria tuberosa extract, rich in antioxidant compounds, alleviates ovariectomized-induced osteoporosis in rats and inhibits growth of breast and ovarian cancer cells". The same terminology has to be applied throughout the text.

Response: The title is changed as: A fraction of Pueraria tuberosa extract, rich in antioxidant compounds, alleviates ovariectomized-induced osteoporosis in rats and inhibits growth of breast and ovarian cancer cells

2. L23: No need to introduce the abbreviation of the genus name, i.e. please delete the bracket. Instead of that, the species authority should be introduced as: Pueraria tuberosa (Roxb. ex Willd.) DC. here and also in the L90. In L89: all the Latin names must stand in italics as well as elsewhere in the text.

Response: Corrections have been made throughout the manuscript (Page: 2, 3, 5, 14, 16-22, 25, 27 & 28).

3. Compound names are randomly capitalized. Please do not write compound names with capital letters unless they present trademarks.

Response: Changes have been made in the manuscript (page: 5 and supplementary material).

4. ARF (antioxidant-rich fraction) cannot stand from the reasons stated as above. "Fraction rich in antioxidant compounds" and a matching abbreviation can read.

Response: ARF (antioxidant-rich fraction) has been stated as "Fraction rich in antioxidant compounds" FRAC throughout the manuscript (Page: 2, 3, 5-7, 9-11, 13, 14, 16-22, 25-28).

5. Figures 8 and 9 are vertically stretched out. Can you revert them to the original ratio?

Response: The figures are now in the original ratio

Reviewer's Responses to Questions

1. If the authors have adequately addressed your comments raised in a previous round of review and you feel that this manuscript is now acceptable for publication, you may indicate that here to bypass the “Comments to the Author” section, enter your conflict of interest statement in the “Confidential to Editor” section, and submit your "Accept" recommendation.

Response:

Reviewer #1: All comments have been addressed

Reviewer #2: (No Response)

Reviewer#2 suggested to report some parameters related to animal experiment. We thank the reviewer and appreciate for the suggestions. We will definitely consider them in our future studies. At this time, it is difficult to conduct the whole animal experiments for only these parameters (redox status in bone and HDL & LDL). We have acknowledged and mentioned them for future studies in the conclusion section. 

2. Is the manuscript technically sound, and do the data support the conclusions?

Reviewer #1: Yes

Reviewer #2: Yes

3. Has the statistical analysis been performed appropriately and rigorously?

Reviewer #1: Yes

Reviewer #2: Yes

4. Have the authors made all data underlying the findings in their manuscript fully available?

Reviewer #1: Yes

Reviewer #2: Yes

5. Is the manuscript presented in an intelligible fashion and written in standard English?

Reviewer #1: Yes

Reviewer #2: Yes

Review Comments to the Author

Reviewer #1: All comments have been addressed. Research ethics was followed as ethical approval was obtained for this study.

We thank the reviewer for encouraging comments.

Reviewer #2: The authors partly responsed my comments. However,

(1) The redox status should be tested in the bone.

(2) The author should explain why did not test HDL and LDL.

Response: We thank the reviewer and appreciate for the suggestions. We will definitely consider them in our future studies. At this time, it is difficult to conduct the whole animal experiments for only these parameters (redox status in bone and HDL & LDL). We have acknowledged and mentioned them for future studies in the conclusion section.

---

## [Editor Report · Decision Letter 2]

26 Dec 2020

A fraction of Pueraria tuberosa extract, rich in antioxidant compounds, alleviates ovariectomized-induced osteoporosis in rats and inhibits growth of breast and ovarian cancer cells

PONE-D-20-26098R2

Dear Dr. Satpathy,

We’re pleased to inform you that your manuscript has been judged scientifically suitable for publication and will be formally accepted for publication once it meets all outstanding technical requirements.

Kind regards,

Branislav T. Šiler, Ph.D.

Academic Editor

PLOS ONE
---

## [Editor Report · Acceptance letter]

4 Jan 2021

PONE-D-20-26098R2 

A fraction of *Pueraria tuberosa* extract, rich in antioxidant compounds, alleviates ovariectomized-induced osteoporosis in rats and inhibits growth of breast and ovarian cancer cells 

Dear Dr. Satpathy:

I'm pleased to inform you that your manuscript has been deemed suitable for publication in PLOS ONE. Congratulations! Your manuscript is now with our production department. 

Kind regards, 

on behalf of

Dr. Branislav T. Šiler 

Academic Editor

PLOS ONE